# Evaluating the Effectiveness of an Augmented Reality Game Promoting Environmental Action

**Kyra Wang**, **Zeynep Duygu Tekler****, Lynette Cheah, Dorien Herremans** * **and Lucienne Blessing**

Information Systems Technology and Design (ISTD), Singapore University of Technology and Design, Singapore 487372, Singapore; kyra_wang@mymail.sutd.edu.sg (K.W.); duygutekler_zeynep@mymail.sutd.edu.sg (Z.D.T.); lynette_cheah@sutd.edu.sg (L.C.); lucienne_blessing@sutd.edu.sg (L.B.)
* Correspondence: dorien_herremans@sutd.edu.sg

**Abstract:** While public awareness of climate change has grown over the years, many people still have misconceptions regarding effective individual environmental action. In this paper, we present a serious game called PEAR, developed using elements of geolocation and augmented reality (AR), aimed at increasing players' awareness of climate change issues and propensity for effective sustainable behaviours. We conducted a study with participants who played the game, gauging their knowledge of and attitudes towards climate change issues before and after playing the game. Our results show that the game significantly improved participants' knowledge on sustainability and climate-change-related issues, and that it also significantly improved their attitudes towards these topics, thus proving that serious games have the potential to impart knowledge and promote sustainable behaviours. Additionally, our results address the lack of empirical studies on the knowledge base of serious sustainability games by introducing methods of quantitatively analysing the effects of serious sustainability games while additionally providing more knowledge about the effectiveness of the specific design elements of our game.

**Keywords:** environmental education; augmented reality; geolocation; mobile games; climate action; gamification; serious games

## 1. Introduction

The climate crisis is becoming a greater threat everyday, as global temperatures march steadily towards unprecedented levels due to human-induced emissions of greenhouse gases. Even with the goals set in the 2015 Paris Agreement, the planet will still reach an average temperature increase of 2.8 °C compared to the 1951 baseline by 2100 [1], which is far above the critical 2 °C tipping point for self-reinforcing temperature-increase feedback in the carbon cycle [2].

Effective climate policymaking depends greatly on citizen support, which is, in turn, dependent on public understanding and awareness of anthropogenic warming [3]. While public awareness of how dire the situation is has steadily increased over the years [4], surveys show that people have major misconceptions regarding individual environmental action, i.e., behaviours they must adopt to be effective at combating climate change [5], and that environmental concern amongst the public has been decreasing globally [6].

A potential aid in correcting this lies in serious games, which have been shown to be able to educate and guide the behaviours of players [7]. Additionally, with the ubiquitous adoption of mobile technologies equipped with advanced data collection and localisation capabilities [8], the mobile games industry has provided developers with opportunities to analyse and reach out to a larger audience than ever before.

Thus, it is our hope that a mobile game with mass appeal can provide a helpful platform to increase climate change awareness and promote sustainable behaviours. In this paper, we present a serious game called PEAR, which was developed using elements of

geolocation and augmented reality (AR) and is aimed at increasing players' awareness of climate change issues and propensity for effective sustainable behaviours. We also provide results of a mixed-method quasi-experimental study establishing the effectiveness of the game at improving players' knowledge and attitudes towards sustainability and climate change issues and intended climate-related behaviours.

In the next section, we will first give an overview of existing evidence showing the effectiveness of games for education and prior work in developing serious games for climate education. We will then outline the design of the developed game. Following that, we then detail a study that we performed with human players to evaluate the effectiveness of using the game to improve the players' attitudes and raise awareness on sustainability and climate change. Finally, we provide the results of the study, our findings, and potential future work.

## 2. Literature Review

Early work on the impact of digital games after their relatively recent rise in popularity over the last 40 years largely focused on their deleterious effects on behaviour, cognition, and affect. These studies had conclusions ranging from violent video games causing increased aggressive thoughts while decreasing pro-social behaviour [9,10] to the addictive nature of video games [11] and the difficulty of regulating time spent playing said games [12].

However, there is a growing body of literature that subverts this idea and shows that it is not all doom and gloom regarding the impact of digital games. The literature suggests that awareness of a subject, while a significant factor [13], is by itself insufficient for achieving behavioural change. According to Michie et al. [14], three essential components—capability, opportunity, and motivation—interact to generate behaviour that, in turn, influences these components. Capability refers to the ability to perform an activity relevant to the behaviour; opportunity refers to factors external to an individual that prompt the behaviour; motivation refers to brain processes that encourage the behaviour. Effective learning thus requires that the learner be motivated, but the motivation must be sustained through feedback, reflection, and active participation [15,16]. While traditional modes of teaching, such as lecturing, are effective for improving learners' recall and comprehension of information, simulation-based learning through computer games is better at engaging learners, enhancing higher-order thinking, and encouraging transfer of actionable skills [17,18].

Recognising the educational potential of video games and gamification, researchers and educators have given more and more attention to serious games, or games designed for a primary goal other than simple entertainment, over recent years [19].

### 2.1. Serious Games for the Climate

Recent studies have proposed that simulations and serious games are uniquely suited for informing, motivating, and changing the attitudes and behaviours of the public towards sustainability issues and challenges [20,21]. Educating people about complex systems such as climate change through textbooks and lectures is a challenging task, as these tools fail to illustrate such systems in a way that actively engages the recipient. Serious games are thus gaining popularity in the field of education and training, as simulation and visualisation technologies allow players to contextualise their experience, thus supporting situated cognition [22].

Games help to educate and engage players in the context of a specific ideological system, where meaning is derived from the experience determined by the game designers [23]. Serious games can thus help to challenge previously held mental models of the world that fail to capture complex realities by providing an opportunity to engage with complex systems through an inhabitable learning system [24]. This attribute makes serious games a particularly suitable vehicle for delivering complex ideas relating to climate change, as experiential learning can help remove previously held misconceptions regarding individual

environmental action. Recognition of this potential has led to the proliferation of serious games targeted at environmental education or addressing sustainability challenges within an educational context [25–27].

Numerous web-based and mobile games already exist to help educate people in various fields of climate change, such as global warming, rising sea-levels, and animal extinction. For example, serious games such as ClimateKids [28] and ClimateChallenge [29] aim to educate young children about the impact of climate change through various tasks and challenges related to greenhouse gas emissions. However, achieving a better public understanding of climate change does not necessarily lead to the desired behavioural change—thus, games such as Greenify [30] use the normative power of social groups to create a culture of positive peer pressure to promote sustainable changes in behaviour. This approach aligns with recent findings investigating the impact of external influences on energy management in workplaces [31]. Yet other games such as PowerAgent [32] take an approach that is more pervasive, utilising real-world data (such as a connection to household electricity meter reading equipment) and everyday activities (such as cooking with a microwave oven instead of an ordinary oven) to blur the boundary between game and real environments, thus increasing the potential for activities in the game to be learned and applied in related non-game activities. A growing body of literature suggests that serious sustainability games are being used in varied settings, ranging from higher education to corporate training, and that there exists an interest in using such tools for addressing sustainability issues due to the perceived relevance of these games and simulations for developing actionable knowledge regarding sustainability [20,33–35].

### 2.2. Distribution of the Existing Knowledge Base on Serious Sustainability Games

Of particular note regarding the existing literature on serious sustainability games is that, according to a bibliometric review of research on simulations and serious games used in educating for sustainability by Hallinger et al. [36], there is a clear imbalance in the distribution of the types of research documents. The knowledge base is heavily skewed towards 'commentaries' (papers that critique existing literature or report on broad trends within the field) at 55% of all reviewed research papers, and it is lacking a critical mass of empirical studies, representing only 33% of the reviewed literature.

Furthermore, these empirical research documents found within the bibliometric review were largely based on non-experimental research designs and descriptive methods. Many authors who claimed to be reporting 'experimental' results were not conducting experiments at all, according to the definition from Campbell and Stanley [37], which states that experimental research design requires a pre-/post-test on relevant variables with a treatment and a control group, where the participants have been randomly selected and assigned. Even though Hallinger et al. [36] utilised a more relaxed operational definition for 'experiments' that did not require random selection and assignment, and even included another 'quasi-experimental' classification that included studies that utilised a pre-/post-test design without a control group, these two less rigorous definitions only encompassed 3% and 6% of the full literature, respectively, with all other non-experimental studies (e.g., cross-sectional surveys, case studies, qualitative studies) representing 24% of the full literature.

This bibliometric review highlights the severe lack of studies in the literature that are able to determine if and how serious sustainability games are capable of achieving the learning outcomes set out by their designers. The unhealthy paucity of empirical studies with strong research designs and methods within the knowledge base on serious sustainability games hinders the field from discovering the true effects of such games and their design elements.

Quasi-experimental studies offer a compromise between experimental and non-experimental studies, as randomisation of participants can sometimes be impractical. An additional challenge in quasi-experimental studies, however, is that estimates of impact are more susceptible to being affected by confounding variables due to the lack of random

assignment, which threatens internal validity, or the truth about inferences regarding causality [38]. Specifically for quasi-experimental studies investigating the impact of serious sustainability games, existing work in the literature that we have reviewed largely does not address the possibility that pre-existing attitudes may have confounding effects when measuring the effectiveness of serious games for knowledge absorption.

One example of a strong research design and method within the literature can be found in a quasi-experimental mixed-method study where Meya and Eisenack [39] used a pre-test/post-test design to provide quantitative evidence on the effectiveness of a simulation game for communicating and teaching international climate politics. Beyond just the basic pre-/post-test design, the authors additionally utilised a chi-square test to compare their results to a nationally representative climate change study, allowing them to make broader conclusions on the generalisability of their results.

Our work in this paper thus aims to fill the identified gap in the literature: Our mixed-method quasi-experimental study both quantitatively and qualitatively analyses the effectiveness of our game in improving the players' knowledge on sustainability and climate change, and we also examine if there is any correlation between the participants' pre- and post-game knowledge. Through this work, we hope to be able to contribute to the growing body of empirical work analysing the effectiveness and impact of serious sustainability games and to develop new techniques for identifying the validity of quasi-experimental research in the field. Before diving into our study design and results, however, we will first describe the developed game.

## 3. The Developed Serious Game—Project PEAR

In this section, we present a mobile game that we developed with the aim of promoting climate action and encouraging sustainable behaviour through a fun and engaging experience. By utilising augmented reality (AR) together with a geolocation function, the game is designed to be educational, engaging, and explorational through an immersive experience. The game is targeted at age groups between 12 to 65 and is set in a post-apocalyptic world. The proposed game is now available in the Google Play Store (https://play.google.com/store/apps/details?id=com.SUTDGameLab.ProjectPear) (accessed on 15 December 2021) and Apple App Store (https://apps.apple.com/sg/app/project-pear/id1504398116) (accessed on 15 December 2021) for free.

### 3.1. Game Design

The game is designed based on a geolocation-based AR mobile game with a story line of a 'Personalised Environmental Assistance Robot' called PEAR (shown in Figure 1). Throughout the game, the robot PEAR tries to revitalise the future Earth from the brink of environmental destruction and tackles various environmental problems during real-world exploration. The exploration feature reflected in the interactive map allows players to collect objects from real locations, which accumulate to a certain point until the environment around the player improves, and the robot PEAR learns the importance of environmental sustainability.

The game was developed using Unity with C♯. The technology used in the game is supported by the current model of smartphones (either Android or iPhones) with the capability of running ARCore and ARkit API, respectively. The geolocation feature relies on the built-in GPS functionality of mobile phones, and is integrated with a list of locations of green buildings in Singapore.

### 3.2. Augmented Reality Pet

As an augmentation of the real world, the game features direct interaction with the robot PEAR, which mimics a pet. Players can increase their affection level when they interact with the robot by rubbing, tapping, and lifting the robot throughout the game. However, the same repetitive interactions do not affect the affection level; therefore, the player must use a combination of these pet interactions to maximise the efficiency of the in-

crement in the affection level. The mechanism behind the interactions with the augmented reality pet is the phone's magnetometer functionality, which detects any electromagnetic fields around electrical appliances in the real world. The software detects and measures the magnitude of the field before translating it to the in-game PEAR charging state.

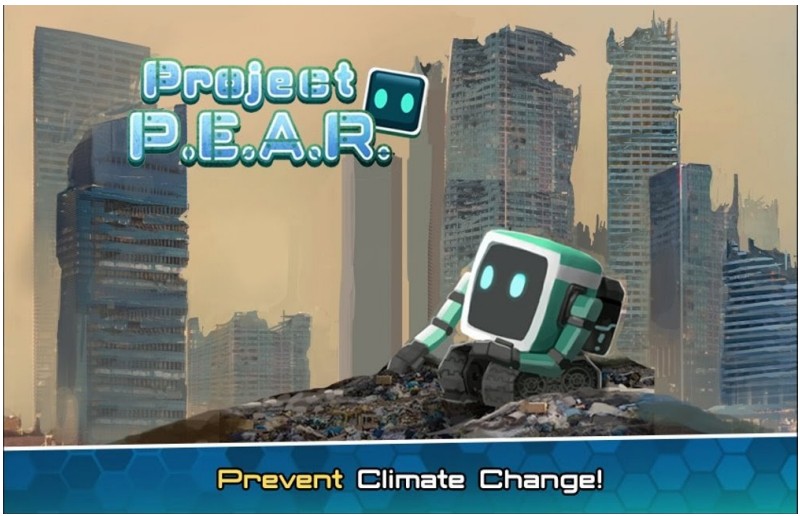

**Figure 1.** Promotional title page for the developed game, Project PEAR.

### 3.3. Map Exploration

Map exploration is the main game feature, which allows players to perform real-world exploration during various quests and mini-games that represent real-world environmental problems. Map exploration is facilitated by the built-in GPS technology of smartphones to support gameplay in the real world. During map exploration, players can see their locations on the map, the location of the environmental problems, and the random trash generated around the player's location (refer to Figure 2). By clearing the trash, they can collect biofuel tokens, which can be exchanged to play various mini-games. Apart from clearing the trash, there are also fixed geoposts around Singapore that are alternative methods for obtaining biofuel tokens. The effort-to-reward balance between geoposts and the trash lies with the amounts of biofuels that are generated as game credits. Players are required to walk towards these geoposts in the real world in order to obtain more biofuel tokens as compensation. Trash is spawned in fixed intervals, thus yielding fewer biofuel tokens as game credits.

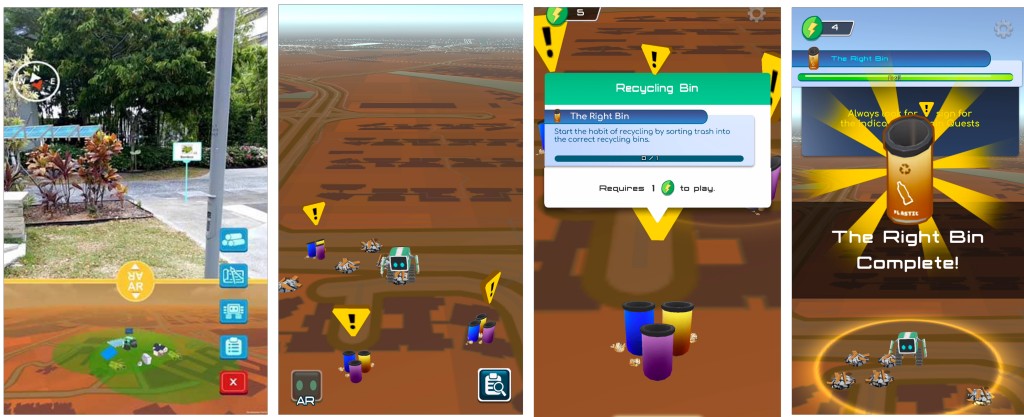

**Figure 2.** Map exploration feature presented in the game.

### 3.4. Mini-Games

There are four different environmental problems to be addressed in each mini-game, namely, recycling, energy conservation, afforestation, and water contamination (shown in

Figure 3). Each mini-game is playable by utilising biofuel tokens as game credits during the map exploration stage, as described in Section 3.3. The player helps the robot PEAR to progress along the quest by clearing the environmental problems through these mini-games. The detailed descriptions of the four mini-games are as follows:

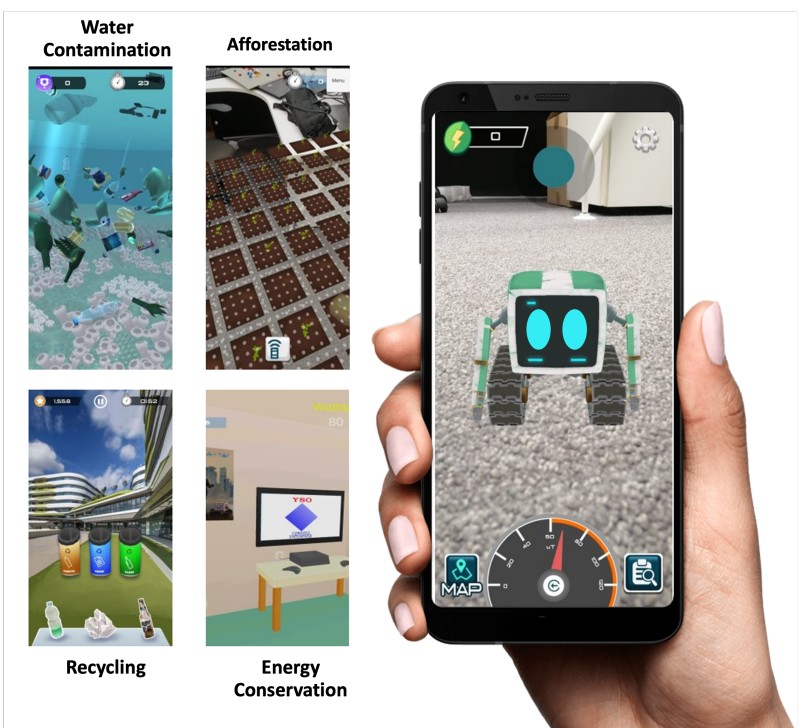

**Figure 3.** User interfaces of the four mini-games and the robot PEAR in augmented reality.

### 3.4.1. Recycling

The Recycling mini-game promotes the behaviour of sorting waste before tossing it into the correct bins to be recycled. The gameplay involves random trash items appearing in front of the players, which need to be sorted into the correct bins. The player is required to identify the waste type—if it is organic, non-organic, or electronic. Correct sorting will be rewarded with points, while incorrect sorting will see points deducted. At the end of the game, a total score is tallied to determine the play performance.

### 3.4.2. Afforestation

The Afforestation mini-game promotes seed planting. The premise of the game is about utilising infrared to scan the soil to reveal soil fertility. The player will need to remember the location of the fertile soil before tapping the screen to plant a seed. Points are awarded for applying seeds on the correct spots, while a penalty is applied when planting seeds on infertile soil.

### *3.5. Energy Conservation*

The Energy Conservation mini-game delivers the message of keeping the electricity usage to a minimum by switching off unused home appliances. There are three different rooms—a bedroom, a living room, and a kitchen—loaded randomly in any given play session. Each room contains different household appliances that the players must search and analyse, and they must switch off any unused appliances in the room.

### *3.6. Water Contamination*

The Water Contamination mini-game delivers the premise of cleaning up the rubbish that has been dumped into rivers or seas. The game puts the player underwater, surrounded by trash. The objective of the game is to clear the trash as much as possible by pointing

the phone towards the surrounding trash in order to collect it. The play performance is measured by the number of trash items collected within the time limit.

## 4. Experimental Setup

### 4.1. Hypotheses

This study hypothesises that players of the game would have stronger intentions to carry out environmentally friendly behaviours (H1) and have greater knowledge about sustainability issues (H2).

**Hypothesis 1 (H1).** *As our game promotes effective sustainable behaviours, players would more likely intend to perform these behaviours after playing the game.*

**Hypothesis 2 (H2).** *As our game imparts knowledge on climate change and sustainability issues, players are more likely to answer knowledge questions on these issues correctly after playing the game.*

### 4.2. Study Design

A mixed method involving a cross-sectional and longitudinal study was performed to evaluate the effectiveness of the developed game in improving players' awareness and knowledge of sustainability and climate change. The study participants were first tasked with answering a pre-game questionnaire in order to obtain their initial attitudes and knowledge on the topic of interest before playing the game. After the participants completed all mini-games, they were tasked with answering a second post-game questionnaire containing a similar set of questions to gauge if there were any changes in the participants' attitudes and knowledge towards sustainability. The study participants were recruited from a local university in the period of January to March 2021. A total of 85 participants completed the pre-game survey, and 37 of those completed all required tasks in the game and the post-game survey.

### 4.3. Questionnaire Design

In this study, the Theory of Planned Behaviour (TRB) Framework [40] was adopted in the design of the questionnaires. The TPB assumes that the intention to perform a behaviour is best predicted when individuals evaluate the behaviour positively (attitudes), believe that their peers will support the behaviour (subjective norm), and perceive the behaviour to be within their capabilities (perceived behavioural control). TPB factors can be assessed directly (e.g., by asking people to report attitudes, norms, and perceived behavioural control) or indirectly (e.g., by asking people about specific behavioural beliefs and combining the scores with a paired evaluation of the belief). In this study, we used a direct assessment through pre- and post-test questionnaires before and after the gameplay.

### 4.4. Behavioural Questions

Both pre- and post-game questionnaires followed a similar structure, where the first section contained a series of behavioural questions designed based on the different factors within the TPB (i.e., attitude, subjective norms, perceived behavioural control) following a 5-point Likert scale, while the second section contained a series of multiple-choice questions to evaluate the participants' knowledge on sustainability and climate change. The details of the assessments for each component of the framework are provided below (see Appendix A).

Attitude: Participants' attitude toward climate change and sustainability while playing the game was assessed.

Subjective norms: The extent to which participants perceived behavioural expectations for climate change and sustainability from people important to them was assessed.

Perceived behavioural control: The extent to which participants perceived that they had control over climate change and sustainability issues was assessed.

Intention to adopt sustainable behaviours: The behavioural intention to adopt sustainable behaviours was assessed.

### 4.5. Knowledge Questions

Knowledge questions were the questions for tracking participants' knowledge on climate change and sustainability. In the pre-game questionnaire, four fact-based multiple-choice questions were asked. In the post-game questionnaire, another four multiple-choice questions were added to understand if they had actually learned the answers throughout the game (see Appendix B).

## 5. Results

### 5.1. Player Retention

An analysis of player data collected from the game's server showed that many players downloaded the game but never even progressed past the introduction, as seen in Figure 4. This could be due to several reasons: The introduction of the game did not engage them enough; they were unwilling to use the game's AR functionality due to the complexity of usage or worries of privacy issues resulting from the requirement of camera permissions; they were unable to use the game any further due to incompatible or insufficient hardware or software capabilities on their device.

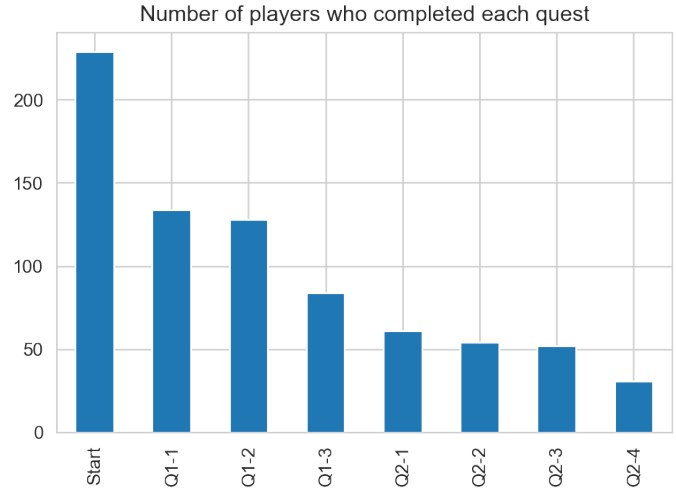

**Figure 4.** Number of players who completed each quest (Q1-1 to Q2-4) in the game, indicating player retention after each quest.

Quests 1-1 and 1-2 are both introductory levels, with simple tasks to familiarise the players with the AR and geolocation functionalities of the game, respectively. Thus, both quests had relatively low completion times, and very few players stopped playing during these quests due to their easy-to-complete nature.

Quest 1-3 is the first quest to be more involved and time-consuming, containing the first mini-game, which focuses on the topic of recycling. As the mini-games require more time and challenge compared to the introductory quests, we saw another drop in player retention compared to quests 1-1 and 1-2, as players were less likely to spend time on a single quest.

Quests 2-1, 2-2, 2-3, and 2-4 are all similar to quest 1-3, as they involve mini-games that are more complex than the introductory activities required in quests 1-1 and 1-2. However, player retention after quest 1-3, the first quest with a mini-game, remained relatively low. This might indicate that players who finished all required quests were inclined to continue playing the game.

### 5.2. Results of the Questionnaires

5.2.1. Behavioural Questions

Figure 5 shows the responses of the participants to the behavioural questions in the pre- and post-game questionnaires. Hypothesis 1 predicted that after playing the game, participants would answer significantly more agreeably to the behavioural questions, indicating that the game improved their intended behaviours towards climate change and sustainability issues.

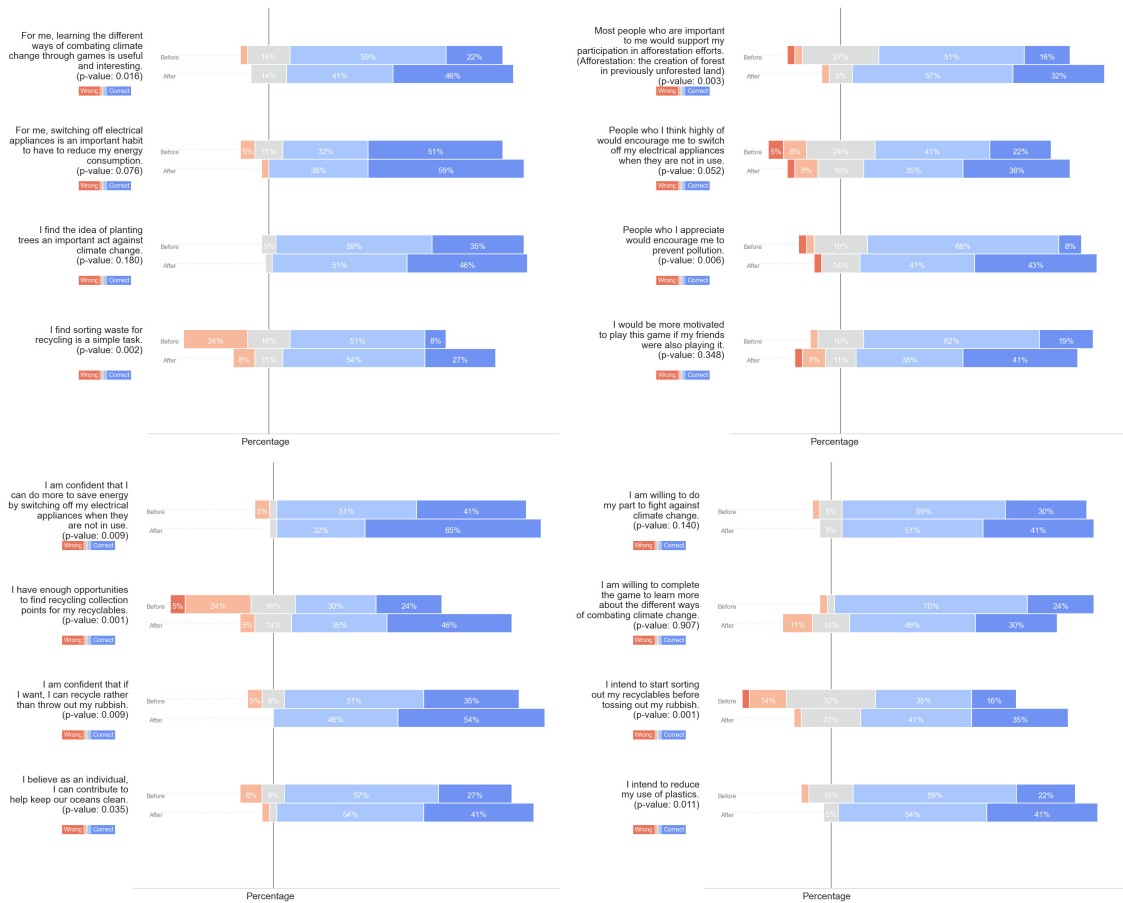

**Figure 5.** Responses from participants for each behavioural question before and after playing the game. Dark red represents "Strongly Disagree" responses, red represents "Disagree" responses, grey represents "Neutral" responses, blue represents "Agree" responses, and dark blue represents "Strongly Disagree" responses. Clockwise, from the top left, the groups of behavioural questions are: attitude, subjective norms, intention to adopt sustainable behaviours, and perceived behavioural control.

A one-sided dependent *t*-test for paired samples was conducted for the results of each behavioural question to test for significance (H0: pre >= post, H1: pre < post, *p* < 0.05). Figure 5 includes the *p*-values for each question.

Significant differences in the pre- and post-game responses were found for some of the behavioural questions, but not all. All four questions assessing perceived behavioural control had significant results, indicating that the game helped participants believe that they had more control over climate change and sustainability than they initially thought.

Questions from other categories specific to certain topics saw significant differences as well, such as recycling ("I find sorting waste for recycling is a simple task": *p* = 0.002; "I intend to start sorting out my recyclables before tossing out my rubbish": *p* = 0.002). These results show that the game possibly helped participants demystify some concepts relating to sustainability, such as the deed of recycling.

For questions that showed no significant difference in pre- and post-game results, many of them already had largely agreeable answers before the game, meaning that there was little to no possibility for an increase in agreement. For example, "I find the idea of planting trees an important act against climate change" had no participant select "disagree" or "strongly disagree" before the game, resulting in an insignificant difference ($p = 0.180$). If participants already largely agreed with a statement before the game, their opinion on the statement after the game was not likely to visibly change, indicating that playing the game had no significant effect on their opinion on the statement or that playing the game confirmed their opinion on the statement.

However, questions related to wanting to play the game itself not only had no significant differences in pre- and post-game results, but had the mean of results actually fall between the pre- and post-game questionnaires ("I am willing to complete the game to learn more about the different ways of combating climate change": $p = 0.907$; "I would be more motivated to play this game if my friends were also playing it": $p = 0.348$). This is in contrast to the significant increase in the general willingness to play games relating to climate change and sustainability ("For me, learning the different ways of combating climate change through games is useful and interesting": $p = 0.016$). This possibly shows that the developed PEAR game specifically did not retain the participants' attention, which is also indicated by the 56.5% participant drop-off after completing the pre-game survey.

### 5.2.2. Knowledge Questions

Hypothesis 2 predicted that after playing the game, participants would answer more knowledge questions correctly, indicating that the game improved their knowledge on climate change and sustainability issues. Figure 6 shows that after playing the game, participants had significantly improved scores on the knowledge questions, regardless of whether the questions were repeated or new.

Figure 6 shows the responses of the participants to the knowledge questions that were repeated in the pre- and post-game questionnaires. While there were generally more correct answers after the game, the questions on recycling and deforestation saw significantly more correct responses, whereas the questions on water pollution and energy did not. This could be due to the types of answers that were provided for each questions; the recycling and deforestation questions had less abstract answer choices (number of mature trees and different measures of area) that were more memorable compared to the water pollution and energy questions, which had more abstract answer choices (billions and weeks/months).

### 5.2.3. Correlation between Behaviour and Knowledge Absorption

As previously noted, quasi-experimental studies suffer threats to internal validity due to the lack of randomisation of participants and the lack of a control group. In the case of this study, as the subjects elected to participate in the study and voluntarily completed all required tasks, self-selection bias might be a confounding factor in the results. Additionally, the participants were recruited from a university campus, a population that does not represent the general populace in terms of pre-existing knowledge and opinions on environmental and sustainability issues. If the effectiveness of our game at improving players' knowledge on sustainability and climate change depends on players' pre-existing attitudes, the study could possibly be a poor representation of the effectiveness of the game in a general populace.

However, within our sample, we found no significant correlation between pre-existing attitudes and the effectiveness of the game at improving players' knowledge and attitudes towards sustainability and climate change, as seen in Figure 7, where the changes in scores on knowledge questions and scores on behavioural questions were normalised by proportionately scaling the change to the amount of potential improvement from the pre-game scores. In fact, there was no significant correlation between attitudinal change and knowledge change before and after the game either. This implies that attitude towards environmental and sustainability issues is potentially independent from knowledge on

environmental and sustainability issues—thus, we are able to treat the two as separate when assessing the efficacy of the game, and pre-existing attitudes do not necessarily affect the efficacy of the game for improving knowledge on environmental and sustainability issues.

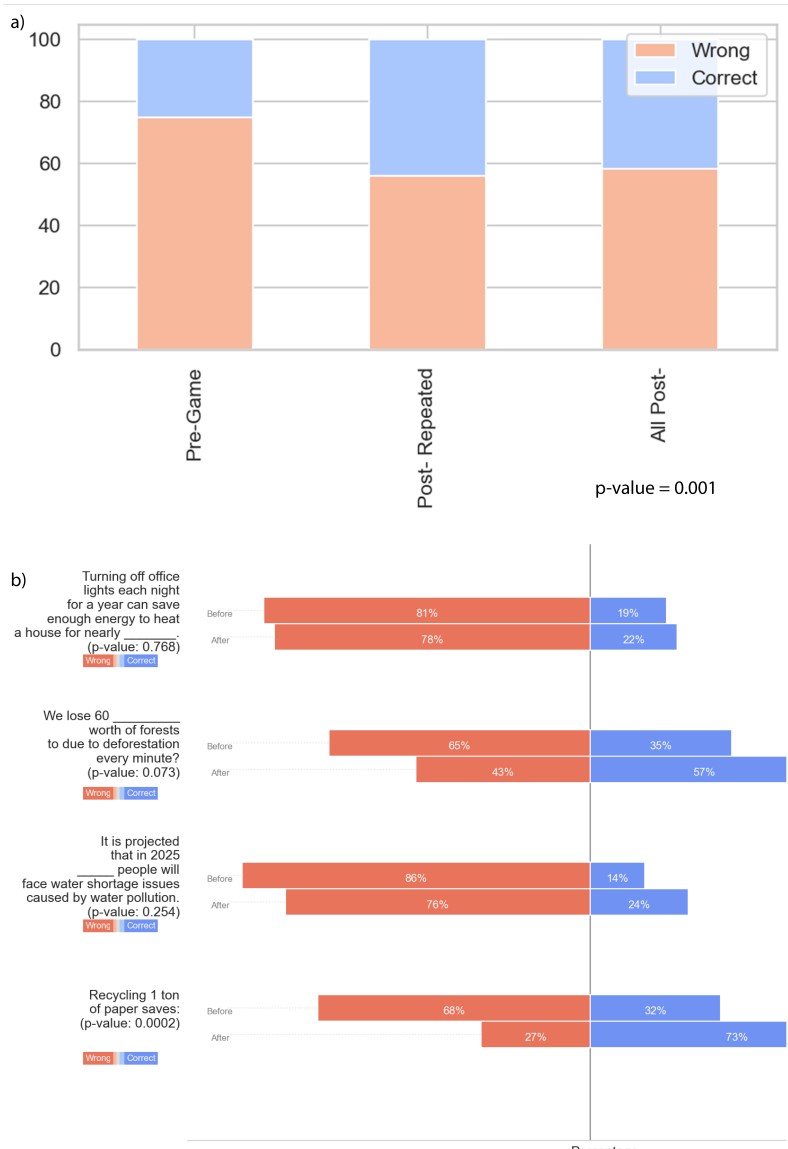

**Figure 6.** Results from the knowledge questions. (**a**) Percentage of correct answers and wrong answers for the knowledge questions. From left to right, the questions represented are: the pre-game questions, the questions in the post-game questionnaire repeated from the pre-game questionnaire, and all of the post-game questions. A one-sided two-sample paired *t*-test where the null hypothesis was that the post-game results were not significantly better than then pre-game results had a *p*-value of 0.001. (**b**) Differences in wrong and correct answers for the questions repeated between the pre-game and post-game questionnaires.

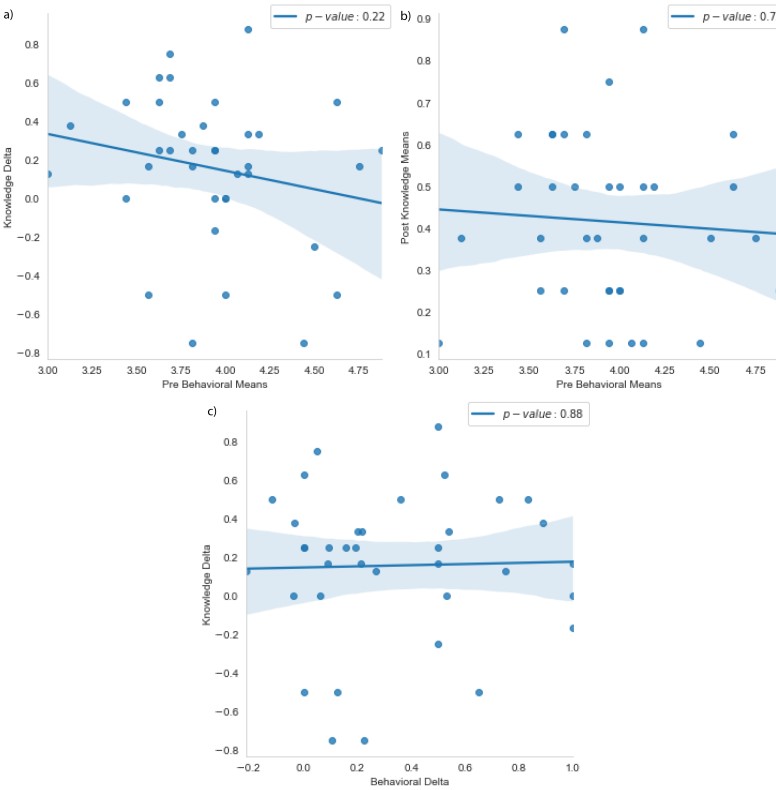

**Figure 7.** Metrics for knowledge absorption plotted against metrics for behaviour. (**a**) Normalised change in knowledge question scores against pre-game mean behavioural question scores. (**b**) Post-game mean knowledge question scores against pre-game mean behavioural question scores. (**c**) Normalised change in knowledge question scores against normalised change in behavioural question scores.

## 6. Discussion

In this paper, we have presented a novel mobile game using elements of geolocation and AR and examined its efficacy in improving players' knowledge and attitudes towards sustainability and climate change. We have shown that the game significantly improved participants' knowledge on sustainability and climate-change-related issues, and that it also significantly improved many related attitudes.

A primary aim of this work was to help address the severe lack of empirical studies within the existing knowledge base of serious sustainability games, as identified by Hallinger et al. [36]. Our mixed-method study quantitatively and qualitatively established the effectiveness of the design of our game. The quasi-experimental nature of our study, however, potentially invites doubts concerning its ability to determine causality between the treatment condition (i.e., playing the game) and the observed outcomes. While the sample of participants is acknowledged to be small and not a true random sample of the general population, we have shown that there does not appear to be a significant correlation between pre-existing attitudes and knowledge absorption during the game, which indicates that self-selection bias and non-random sampling did not have too great of an effect on the results regarding knowledge absorption. Beyond this result, this work also provides a novel method for determining the generalisability of future quasi-experiments conducted within the serious sustainability game space. However, future studies could improve on this anyway by randomly sampling from the general population, ensuring that all selected participants complete the game, and establishing a control group in order to achieve a truly experimental research design.

We acknowledge that a limitation of our questionnaire design is that it only measures self-reported attitudes related to sustainability, which do not necessarily reflect the practical changes and implications in the activities and behaviours of players or their actual impact

on the environment. Additionally, levels of agreement were not universal between participants. The participants' actions may not accurately reflect their survey answers, and one participant's "strongly agree" might be equivalent to another's "neutral". Future studies could track participants' actions over a period of time before and after the game, such as with the frequency of recycling activities, to more accurately show the effectiveness of the game in a practical sense. Additionally, another potentially interesting line of investigation could be to measure if there are significant disparities between the self-reported attitudes of participants and their actual changes in sustainable behaviours, which would indicate resistance in the subjects in actually making practical changes to their lifestyles.

A shortcoming of the game appears to be its inability to engage and retain players. A significant reduction in the number of active players was observed as they progressed through the game. As effective learning requires that the learner is motivated to participate in the activity [15], future changes in the game design should aim to improve this aspect, and future studies could focus on polling participants on how to improve the game's engagement. It has been shown that increasing the challenge of a game has positive effects on learning both directly and via increased engagement [41], and thus, we could also consider deepening the content and challenges provided within our game in order to make playing the game an intrinsically interesting activity, invoking a flow experience.

**Author Contributions:** Conceptualisation, L.C., D.H. and L.B.; methodology, all; validation, K.W. and Z.D.T.; formal analysis, K.W. and Z.D.T.; investigation, all; resources, L.C., D.H. and L.B.; data curation, K.W. and Z.D.T.; writing—original draft preparation, K.W. and Z.D.T.; writing—review and editing, all; visualisation, K.W.; supervision, L.C., D.H. and L.B.; project administration, L.C., D.H. and L.B.; funding acquisition, L.C., D.H. and L.B. All authors have read and agreed to the published version of the manuscript.

**Funding:** This research was funded by the SUTD-MIT International Design Centre.

**Institutional Review Board Statement:** The study was conducted according to the guidelines of the Declaration of Helsinki and approved by the Institutional Review Board of the Singapore University of Technology and Design (protocol code 20-00380, approved on 5 January 2021).

**Informed Consent Statement:** Informed consent was obtained from all subjects involved in the study.

**Acknowledgments:** We thank Yogabrata Datu and Danny Chow from the SUTD Game Lab for their role in game design and development of Project PEAR.

**Conflicts of Interest:** The authors declare no conflict of interest. The funder had no role in the design of the study; in the collection, analyses, or interpretation of data; in the writing of the manuscript, or in the decision to publish the results.

## Appendix A. Behavioral Questions

| Behavioral Questions |
| --- |
| **1. Attitude** |
| I find sorting waste for recycling is a simple task. |
| I find the idea of planting trees an important act against climate change. |
| For me, switching off electrical appliances is an important habit to have to reduce my energy consumption. |
| For me, learning the different ways of combating climate change through games is useful and interesting. |

**2. Subjective Norms**

I am more motivated to play this game because my friends are playing it as well.
People who I appreciate would encourage me to prevent pollution.
People who I think highly of would encourage me to switch off my electrical appliances when they are not in use.
Most people who are important to me would support my participation in afforestation efforts.

**3. Perceived Behavioural Control**

I believe as an individual, I can contribute to help keep our oceans clean.
I am confident that if I want, I can recycle rather than throw out my rubbish.
I have enough opportunities to find recycling collection points for my recyclables.
I am confident that I can do more to save energy by switching off my electrical appliances when they are not in use.

**4. Behavioural Intention**

I intend to reduce my use of plastics.
I intend to start sorting out my recyclables before tossing out my rubbish.
I am willing to complete the game to learn more about the different ways of combating climate change.
I am willing to do my part to fight against climate change.

**Appendix B. Knowledge Questions**

**Knowledge Questions**

**Recycling**

Pre–Post: Recycling 1 ton of paper saves:
- 3 mature trees
- 5 mature trees
- 10 mature trees
- 17 mature trees
Post: What percentage of glass is estimated to be recycled into new containers?
- 10%
- 30%
- 50%
- 80%

**Water Contamination**

Pre–Post: The most common type of waste found in oceans is:
- Diapers
- Food wrappers
- Cigarette Butts
- Plastic bottles
Post: How many people will face water storage issues due to water pollution in 2025?
- 1 million
- 3.5 million
- 5 million
- 10 million

| **Afforestation** |
| --- |
| Pre–Post: Four primary drivers of tree loss are beef, soy, palm oil, __________<br>- Timber<br>- Rubber<br>- Olive<br>- Tea Leaves<br>Post: We lose 60 _________ worth of forests to deforestation every minute.<br>- Football Fields<br>- Olympic Swimming Pools<br>- Square Meters<br>- Hectares |
| **Energy Conservation** |
| Pre–Post: Laptops use up to ______ less electricity than desktop PCs.<br>- 15%<br>- 30%<br>- 65%<br>- 85%<br>Post: Turning off office lights each night for a year can save enough energy to heat a house for nearly _______.<br>- 1 week<br>- 6 weeks<br>- 3 months<br>- 5 months |

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
