# Peer review of "Evaluating the Effectiveness of an Augmented Reality Game Promoting Environmental Action"

_sustainability, doi:10.3390/su132413912_

Round 1
Reviewer 1 Report
The intentions and research questions for this study are certainly admirable, and I appreciate the work that went into developing the game.
I encourage you to conduct additional research in order to investigate any possible resistance within the subjects in actually making changes to knowledge or behavior in climate change. You might also consider deepening the content and challenges within the game.
Author Response
1. I encourage you to conduct additional research in order to investigate any possible resistance within the subjects in actually making changes to knowledge or behavior in climate change.
This point has been added as possible future work in the discussions section.
2. You might also consider deepening the content and challenges within the game.
This point has been added as possible future work in the discussions section.
Reviewer 2 Report
Reviewer Report
Paper: Evaluating the Effectiveness of an Augmented Reality Game Promoting Environmental Action
This paper present a serious game called PEAR, developed using elements of geolocation and augmented reality (AR), aimed at increasing players’ awareness of climate change issues and propensity for effective sustainable behaviours. The topic is very interesting and of great current relevance.
The following are some recommendations to enhance the study.
MAIN CONCERN:
1.Update the literature and include contemporary studies, especially from the last two or three years
2.Establish the relevance of the study
3.Expand the discussion section
4.Include a conclusions section
5. Authors should check the keywords: one of them (environmental education) is only included in the abstract.
COMMENTS
Line 94 The study by Tragazikis and Meimaris Tragazikis and Meimaris .
It should be: The study by Tragazikis and Meimaris
Author Response
1. Update the literature and include contemporary studies, especially from the last two or three years
The literature review has been reworked to include more contemporary studies, including several published as recently as 2020.
2. Establish the relevance of the study
The reworked literature review references a 2020 bibliometric review of sustainability serious games that identifies a severe lack of empirical studies within the existing knowledge base. Our mixed-method study thus fills a glaring gap in the literature by introducing methods of quantitatively analysing the effects of sustainability serious games, while additionally providing more knowledge about the effectiveness of the specific design elements of our game.
3. Expand the discussion section
We have expanded the discussion section to include elaborations of: our fulfilment of our goal to fill the identified lack of empirical studies within the existing knowledge base; our acknowledgement that while the study does not measure practical impact, the game’s aim beyond the study is indeed to promote positive practical changes and implications to the activities and behaviours of players, and that future work can utilise metrics to measure such practical impact; and our intended future work in increasing the challenge of the game to invoke a flow experience and measuring the disparity between self-reported attitudes and actual behavioural changes.
4. Include a conclusions section
According to the MDPI Instructions for Authors (https://www.mdpi.com/journal/information/instructions), the conclusions section is not mandatory, be can be added to the manuscript if the discussion is unusually long or complex. We do not believe that this is this case, and thus have opted to not include a conclusions section to avoid unnecessary rehashing of points already brought up in the discussions section.
5. Authors should check the keywords: one of them (environmental education) is only included in the abstract.
We have included references to all keywords in the main body of the paper as well.
Reviewer 3 Report
Since the authors acknowledge that the "questionnaire design is that it only measures intended be haviours, and not actual behaviours", the whole text shoul be re-written in order to make it sensible, from there resulting a narrative of findings from EITHER participants' behaviours OR participants' intended behaviours, since this makes ALL the difference.
Author Response
1. Since the authors acknowledge that the "questionnaire design is that it only measures intended be haviours, and not actual behaviours", the whole text shoul be re-written in order to make it sensible, from there resulting a narrative of findings from EITHER participants' behaviours OR participants' intended behaviours, since this makes ALL the difference.
We have made it more explicit in the paper that the study aims only to measure the effectiveness of the game at improving players' knowledge and attitudes towards sustainability and climate change issues, and intended climate-related behaviours. We also have made it clear that while the study does not measure practical impact, the game’s aim beyond the study is indeed to promote positive practical changes and implications to the activities and behaviours of players, and that future work can utilise metrics to measure such practical impact.
Round 2
Reviewer 1 Report
This revision reflects a good deal of the comments raised by reviewers, which I think has improved the paper.
The bigger question is about the efficacy or intended outcome of the game itself in terms of its intended purposes. Future research will have to tell the rest of that story.
Reviewer 3 Report
I feel I have to maintain my previous review, since the identified contradiction persists.